ecology, behaviour

agriculture, hoverfly, migration, pollination, Syrphidae, insect declines

**Author for correspondence:**
Karl R. Wotton
e-mail: k.r.wotton@exeter.ac.uk

# Pollination by hoverflies in the Anthropocene

Toby Doyle[1], Will L. S. Hawkes[1], Richard Massy[1], Gary D. Powney[2,3], Myles H. M. Menz[4,5,6,7] and Karl R. Wotton[1]

[1]Centre for Ecology and Conservation, University of Exeter, Cornwall Campus, Penryn, UK
[2]UK Centre for Ecology and Hydrology, Benson Lane, Crowmarsh Gifford, Wallingford OX10 8BB, UK
[3]Oxford Martin School and School of Geography and Environment, University of Oxford, Oxford, OX1 3BD, UK
[4]Department of Migration, Max Planck Institute of Animal Behavior, Radolfzell, Germany
[5]Centre for the Advanced Study of Collective Behaviour, and [6]Department of Biology, University of Konstanz, Konstanz, Germany
[7]School of Biological Sciences, The University of Western Australia, Crawley, Western Australia, Australia

GDP, 0000-0003-3313-7786; MHMM, 0000-0002-3347-5411; KRW, 0000-0002-8672-9948

Pollinator declines, changes in land use and climate-induced shifts in phenology have the potential to seriously affect ecosystem function and food security by disrupting pollination services provided by insects. Much of the current research focuses on bees, or groups other insects together as 'non-bee pollinators', obscuring the relative contribution of this diverse group of organisms. Prominent among the 'non-bee pollinators' are the hoverflies, known to visit at least 72% of global food crops, which we estimate to be worth around US$300 billion per year, together with over 70% of animal pollinated wildflowers. In addition, hoverflies provide ecosystem functions not seen in bees, such as crop protection from pests, recycling of organic matter and long-distance pollen transfer. Migratory species, in particular, can be hugely abundant and unlike many insect pollinators, do not yet appear to be in serious decline. In this review, we contrast the roles of hoverflies and bees as pollinators, discuss the need for research and monitoring of different pollinator responses to anthropogenic change and examine emerging research into large populations of migratory hoverflies, the threats they face and how they might be used to improve sustainable agriculture.

## 1. Introduction

Animal-mediated pollination is a critical process for supporting both natural ecosystems and human food security by facilitating reproduction of much of the world's plant life [1]. Of those plants selected as crops by humans over the last 13 000 years, around 76% are dependent on animal pollination [2,3]. The gross economic value of the 105 most widely planted crops that are pollinated by insects amounts to more than US$800 billion per annum [4], while the value to native plant species is immeasurable. Wild and managed bees are widely regarded as the most important group of pollinators for crops and have been studied extensively [5–7], but there is growing interest in the role of 'non-bee insects' as pollinators [4,8–10]. Of these other insect groups, hoverflies (Diptera: Syrphidae) have emerged as the most prominent pollinator taxa.

The family Syrphidae, also called hoverflies, flower flies or syrphid flies, is made up of approximately 6000 species in around 200 genera and occur on every continent except Antarctica and remote oceanic islands [11,12]. The family is traditionally organized into three subfamilies (though see [13]), two of which are considered particularly important in terms of pollination, the Syrphinae and the Eristalinae, made up of around 1800 and 3800 species, respectively [11]. By contrast, the approximately 400 species of Microdontinae often do not rely on flowers as adults. Diversity within the Syrphidae is dramatic and wide ranging at both the larval and adult stages (figure 1). Larval feeding modes

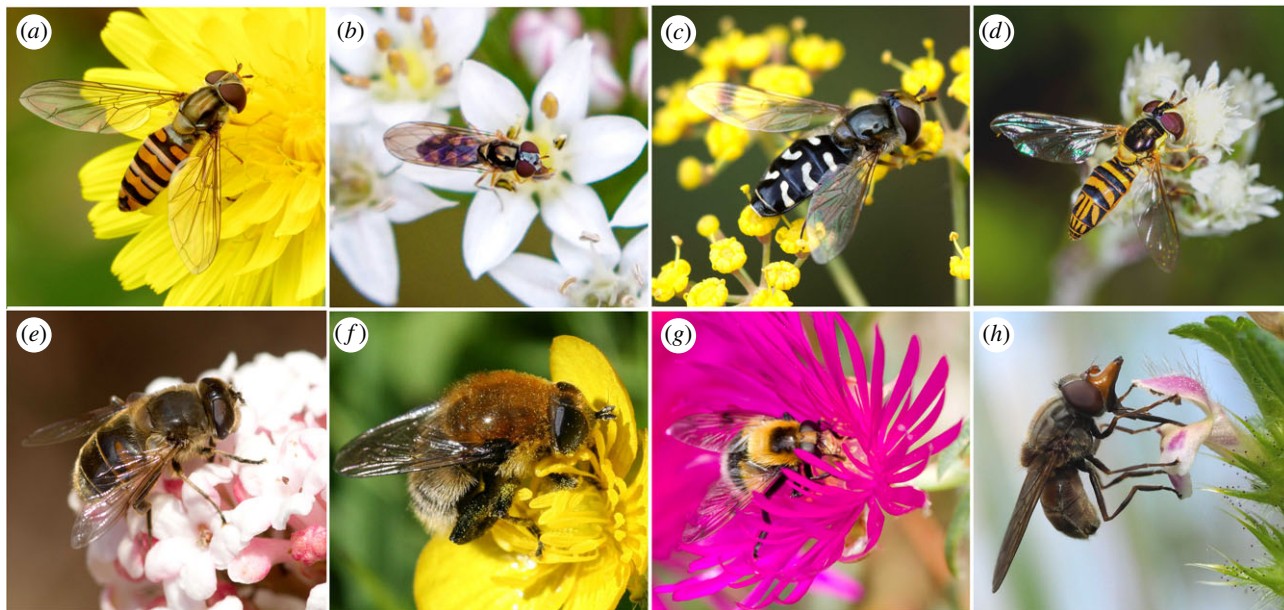

**Figure 1.** Hoverfly diversity. Selected members of the subfamilies Syrphinae (*a–d*) and Eristalinae (*e–h*) mentioned in the text. (*a*) The marmalade hoverfly *Episyrphus balteatus*—a distinctive and highly migratory hoverfly widespread throughout the Palaearctic region. (*b*) The variable duskyface *Melanostoma mellinum*—a small abundant species found throughout the Palaearctic North Africa and North America known to be migratory in Europe. (*c*) The pied hoverfly *Scaeva pyrastri*—widespread and highly migratory. (*d*) The oblique stripetail *Allograpta obliqua*—a common North American species. (*e*) The dronefly *Eristalis tenax*—a cosmopolitan honeybee mimic and highly migratory at least in Europe and the East Coast of North America. (*f*) The narcissus bulb fly *Merodon equestris*—a polymorphic bumble bee mimic found in the Holarctic region. Larvae feed internally in tissues of bulbs. Introduced to New Zealand and believed to have been introduced into Britain from Europe in daffodil bulbs at the end of the nineteenth century. [14]. (*g*) The bumblebee hoverfly *Volucella bombylans*—a large polymorphic bumblebee mimic (yellow form pictured) found in the Palaearctic and Nearctic regions. Larvae live in the nests of social wasps or bumblebees. (*h*) The Heineken fly *Rhingia campestris*—a distinctive snout hides a long proboscis used for feeding on deeper flowers. Found throughout the Palaearctic region its larvae are associate with cattle dung but it may live in other wet media. (*a*) By Katja Schulz CC BY 4.0. (*b,d*) By Melissa McMasters and bob15noble, respectively, CC BY-NC 4.0. (*c,e,g*) By Will George (*h*) by Frank Vassen CC BY-NC 2.0. (*f*) By S. Rae CC BY 2.0. All images have been cropped and adjusted. (Online version in colour.)

include various forms of zoophagy, phytophagy, coprophagy and saprophagy (see [11,15] for a full account). In many cases, these stages provide important additional ecosystem services, for example, some species in the subfamily Syrphinae provide biocontrol of crop pests, consuming vast numbers of aphids during their development [11,15–19] while filter-feeding saprophagous larvae are very common among the Eristalinae and include 'rat-tailed maggot' forms adapted to aquatic environments and important for the recycling of waste [11,20]. Adults typically feed on nectar and pollen and their morphologies range from large, hirsute bumblebee mimics, to miniscule hairless species with mimicry of bees and wasps widespread [21,22]. Given their important role in pollination and provision of other ecosystems functions, hoverflies are gaining particular interest as beneficial species and alternative managed pollinators. Here, we provide a synthesis of hoverflies as pollinators, highlight the impact of anthropogenic change on their populations and discuss key avenues for future research.

## 2. Hoverflies as pollinators

Bees provision their young with nectar and pollen, and in the case of social bees, may operate at a very high density to achieve this. By contrast, flowers are vital to hoverflies for a different reason: they provide nectar as a food source and the pollen required for ovarian development [23]. This distinction is important as hoverflies are not restricted to a limited home range and may carry pollen over longer distances than bees while foraging [24–26] (figure 2*a*), and over considerably longer distances during migration [16]. Some hoverflies may

also be present in very high densities, which is particularly true of migratory species. Radar studies of two common European species, *Episyrphus balteatus* (figure 1*a*) and *Eupeodes corollae* estimated up to 4 billion individuals move over southern Britain each year (figure 2*b*). Such numbers rival the 5 billion managed honeybees at peak abundance for the whole of Britain [16]. In and around agroecosystems, hoverflies are often many times more abundant than all wild bee species, and this natural abundance may make up for potentially lower pollinator efficiency [27–29].

Recently, Rader *et al.* [4] investigated the relative importance of crop pollinators, including hoverflies, using reports of insect visitors to 105 global crop plants [4]. The study revealed Diptera (true flies) as the second most important order of pollinating insects, visiting 72% of crops compared to 93% for Hymenoptera (bees, wasps and ants) and 54% for Lepidoptera (butterflies and moths). Among the Diptera, hoverflies visited 52% of these crops, which we estimate to be worth a gross economic value of around US$300 billion per year based on data from the Food and Agriculture Organisation for 2017 [30]. Visitation to these crops by hoverflies was only surpassed by the bee families Apidae, which includes honeybees and bumblebees (90% of crops), and the sweat bees Halictidae (58% of crops) (figure 2*c*). Two hoverfly species, *E. balteatus* and *Eristalis tenax* (figure 1*a* and *e*), had the highest visitation rates of 24 and 28 different crop plants, respectively. However, given the relative lack of investigation into pollination of agricultural crops by hoverflies, these values are likely to be significant underestimates.

Hoverflies are important pollinators of wildflowers in many ecosystems [8,9,31,32]. In Europe, hoverflies have been

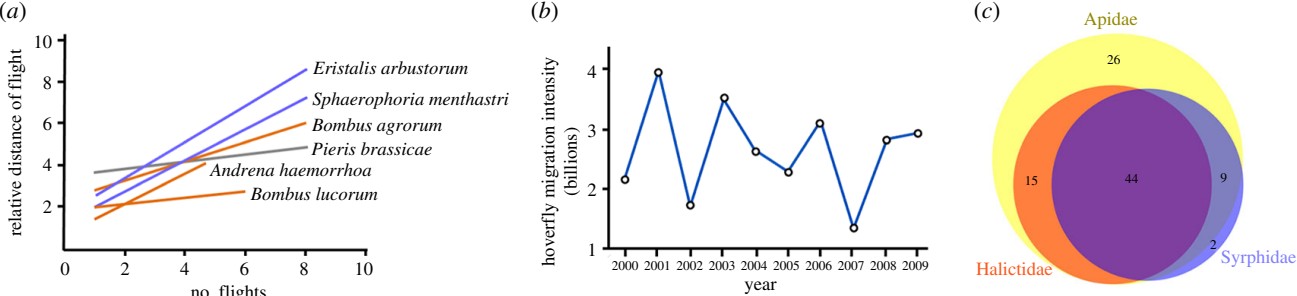

**Figure 2.** Hoverfly foraging distances, abundance and crop visitation. (*a*) Hoverflies may carry pollen over longer distances while foraging than bees and butterflies. Flight distance constructed as regression lines of the relative distance (flight length/mean distance between recording units of flowers, inflorescences or plants) plotted against the number of flights for hoverflies (blue lines) bees (orange lines) and butterflies (grey line). Redrawn from [25]. (*b*) Annual totals of the migrant hoverflies *Episyrphus balteatus* and *Eupeodes corollae* in billions over southern Britain during a 10-year period. Redrawn from [16]. (*c*) The top three most frequent insect visitors by family to 105 global crop plants. Plots made using data in [4]. (Online version in colour.)

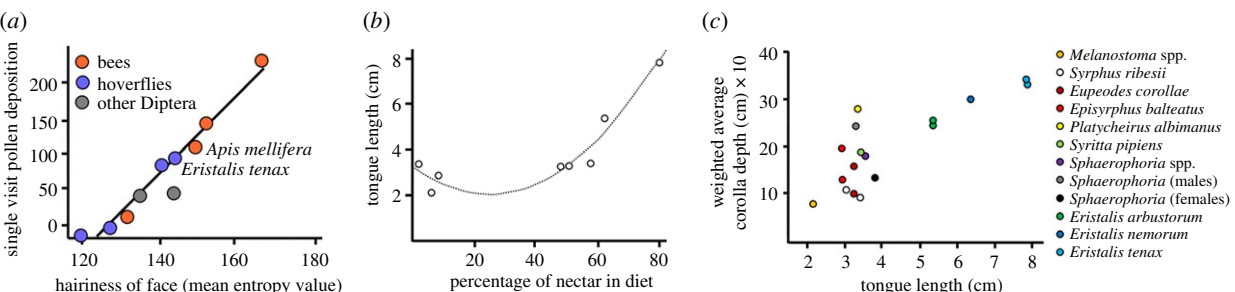

**Figure 3.** Hoverfly morphology and pollinator effectiveness. (*a*) Relationships between hairiness of the face and single visit pollen deposition for hoverflies (blue) bees (orange) and other Diptera (grey) highlighting the comparable effectiveness of honeybees and *Eristalis tenax* on *Brassica rapa*. It is not currently known if effectiveness is also comparable between the bumblebee *Bombus terrestris* (top right data point) and hoverfly bumblebee mimics. Redrawn from [43]. (*b,c*) Relationships between average tongue length in different species of hoverflies and percentage nectar in the diet (*b*) or average flower corolla depth (*c*). As tongue length increases the proportion of pollen in the diet decreases and hoverflies concentrate on flowers with longer corollas. Redrawn from [31]. (Online version in colour.)

found to visit more than 70% of animal-pollinated wildflower species [11]. An investigation into pollen transport networks in conservation grasslands showed that *Eristalis* hoverflies may transport the pollen from 65 plant taxa, higher than previously reported from flower observations alone [33]. This study also showed generalization in flower visitation at the species level but a degree of floral consistency at the individual level [33,34]. Floral consistency is an important factor for pollinator efficiency and has previously been documented for two other species of hoverflies, *E. balteatus* and *Syrphus ribesii*, foraging on wildflowers [35]. Some hoverflies have also been shown to exhibit innate colour preferences, for example, *E. tenax* and *E. balteatus* show a strong preference for yellow flowers [36,37] and *Volucella bombylans* and *Rhingia campestris* (figure 1*h,g*) for blue flowers [38,39]. In addition, *E. tenax* has been shown to visit the ring florets on capitulum flowers systematically, leaving once a full circle has been completed [40]. Finally, there exist a number of specialist interactions between various species of orchid and hoverflies [41,42] and other such interactions may await discovery.

Pollinator efficiency can be influenced by size and morphology. These characteristics determine the depth at which the hoverfly can forage for nectar and the quantity of pollen that can be carried on the hoverflies' body. Hairiness has been shown to be a good predictor of pollen load and pollinator efficiency based on the number of conspecific pollen grains deposited on a virgin stigma in a single visit (figure 3*a*) [43]. However, relative effectiveness in terms of pollen deposition will vary depending on the plant species involved. In New Zealand, *E. tenax* has been shown to transfer a similar

numbers of pollen grains as honeybees to the stigmas of pak choi and onions [29,44], but to be less effective on kiwifruit [43]. In apple orchards, hairy bumblebee mimics like *Merodon equestris* (figure 1*f*) can carry around 10 000 pollen grains (of which 29% was fruit pollen) while honeybee mimics such as *E. tenax* can carry around 3500 (67% fruit pollen) [45]. By contrast, pollen loads of bumblebees *Bombus terrestris* and honeybees *Apis mellifera* were 19 000 (85% fruit pollen) and 5600 (73% fruit pollen), respectively [45]. Mouth parts vary widely between hoverflies. The long snout of *R. campestris* (figure 1*h*) encloses a proboscis of over 10 mm in length allowing it to visit flowers with deep tubes [46]. The longer proboscis of eristaline species such as *R. campestris*, *E. tenax* (7.85 mm) and *V. bombylans* (7.24 mm), together with smaller labella, are correlated with increased nectar feeding on deeper, narrower flowers (figure 3*b,c*) [31,46]. *Episyrphus balteatus*, with a tongue of 2.9 mm, is typical of the shorter proboscides seen in the Syrphinae, which together with proportionally larger labella, are well suited for feeding on shallow, open flowers [11,46].

Pollinator efficiency, as measured by seed set following a single visit from a potential pollinator to a virgin flower, has seldom been quantified for non-bee flower visitors. However, seed set has been measured following multiple visits for various hoverfly species. In semi-field experiments, Fontaine *et al.* [47] showed that in plant communities containing only open flowers, plants produced a significantly higher mean number of seeds per fruit when visited by hoverflies compared to bumblebees. The authors concluded that bumblebees were less-efficient pollinators than hoverflies of open flowers [47].

Further evidence of potential hoverfly pollination efficiency has been demonstrated through improved seed set in greenhouse sweet peppers of between 9% and 19% when visited by *E. tenax* over non-visited control groups [48], while the suggested yield gains in oilseed rape (*Brassica rapa*) were 15–25% when pollinated by *E. balteatus* over controls [49], and 70% for strawberry yields in a mixed hoverfly semi-field experiment [50]. Further work by Jauker *et al.* [51] also revealed increasing yields of oilseed rape with increasing pollinator densities, with a mix of *E. balteatus* and *E. tenax* (96 per 7.5 m$^2$) resulting in yields close to those achieved by small honeybee colonies (200 per 7.5 m$^2$) [51].

Interactions between hoverfly species, and with other pollinators, has been understudied but is likely to be important in terms of competition for resources [52,53]. Territoriality is common among male hoverflies. For example, male *E. tenax* and *M. equestris* aggressively defend patches of flowers (typically 1–2 m$^2$ for *E. tenax*) from conspecifics, but also from other flying insects, though this appears to be restricted to summer generations [54]. This chasing and striking behaviour can lead to serious injury to hoverflies and bees, including death from broken necks, and may also prevent pollination in the defended territory for long periods of time [52]. The secondary effects of this behaviour for the provision of ecosystem services in the surrounding areas has yet to be investigated.

## 3. Hoverfly migration and long-distance pollen transfer

Unlike bees, which rarely forage over distances exceeding 1–2 km [55–58] (though this may increase depending on foraging conditions [59]), many species of hoverflies are highly migratory [60,61]. Hoverfly migration is known from North America [62,63], Asia (Nepal: [64]) and Australia [65,66], but is best understood in Europe where seasonal influxes into northern regions begin around May and are followed by often huge southwards migrations during August–October [16,60,61,67]. During this southward journey, hoverflies may cover hundreds of kilometres in a single day [16,68] and thousands of kilometres over the entire period [69]. Migration provides significant reproductive advantages, with subsequent populations of *E. balteatus* and *Eupeodes* spp. reaching an average of 4.5 times those entering the UK in Spring [16].

Migrating hoverflies are capable of transporting pollen over long distances including greater than 100 km over open water (WLSH 2019, personal observation). Although pollen loads on some migrating hoverflies have been observed to be lower than those collected from agricultural areas, migrating flies can still transport billions of pollen grains [16]. As pollen carried by migrating insects can stay viable for up to 2 days [70], hoverflies will be capable of transporting viable pollen over long-distances, thereby facilitating high levels of gene flow between plant populations that would otherwise remain unconnected. This, in turn, may have beneficial consequences for plant population health and fruit yield [71], and secondary benefits for non-migratory pollinators that may visit the same plant species. Migratory pollinators may also be particularly important for geographically isolated plant populations where a lack of local pollinators limits pollen transfer. For example, Pérez-Bañón *et al.* [72,73] demonstrated the importance of long-distance migratory pollinators on the Columbretes archipelago (Mediterranean Sea) where bees are

absent and pollination is carried out primarily by the migratory *E. tenax* [72,73]. Under such conditions, migratory pollinators may support the persistence of some isolated plant species and their conservation may require the protection of source areas on the coast [73]. Indeed, the threats facing migrant hoverflies in transit have not been investigated, but landscape connectivity is likely to be an important factor. Other migratory pollinators use corridors made up of populations of sequentially blooming plants and populations eliminated from this sequence may have major consequences for the movement capabilities and survival of migratory species [74].

## 4. Encouraging hoverflies in agricultural ecosystems

Hoverfly species richness and abundance increases in complex agricultural landscapes incorporating features that provide a temporally stable supply of resources such as food, shelter and larval habitat [24,75,76]. Woody elements such as hedgerows are associated with increased local abundance of hoverflies and ecosystem services such as biocontrol of crop pests [77,78], especially when connected to forest [79]. The quantity of grassland habitat in a landscape has also been shown to scale positively with hoverfly diversity [24,80,81], while management practices such as delayed mowing have been shown to increase abundance by providing foraging resources for longer in the season [82]. To capitalise on the pollination (and often pest-control) services they provide, measures are being increasingly implemented to encourage hoverflies and other beneficial species into agricultural systems, such as planting of flower strips [83]. Nectar accessibility is the dominant factor in flower choice for *E. balteatus* and the abundance of accessible flowers is positively correlated with zoophagous hoverfly numbers [84]. In Europe, ideal seed mixes for hoverflies feature large flat inflorescences such as Umbelliferae [85], with only a few key plant species required to attract the majority of common species [86]. Simplified agricultural landscapes lacking in diverse habitat benefit the most from improved resources, increasing hoverfly abundance with spill-over into adjacent fields [79,87]. Habitat demands change between guilds, with ponds and aquatic features essential for non-aphidophagous hoverflies with aquatic larvae such as *E. tenax*, increasing their abundance, related pollination services and fruit yield [88]. Conservation agricultural practices such as crop multi-culture can provide resources for both larval and adult hoverflies [89] and mitigate the effects of landscape simplification by maintaining in-field habitat [90,91]. Reduced soil tillage and the presence of woody habitats such as hedgerows leaves overwintering habitat for hoverflies that can boost important early season ecosystem services such as biocontrol of crop pests [90].

Hoverflies persist predominantly as wild pollinators with no managed systems employed to the degree of managed honeybees or commercially reared bumblebees, though commercial rearing is on the rise. Pineda *et al.* [92] discovered large populations of hoverflies operating in greenhouses at high temperature during spring and summer in south-eastern Spain, together with a temporal succession of species during the growing season ending with *Sphaerophoria rueppellii* the seemingly most adapted to survival at higher temperatures and drier conditions [92]. Consequently, hoverflies show huge potential to be incorporated as managed pollinators in

many agricultural systems. In a few cases, hoverflies appear to have been introduced into countries to aid pollination. *Simosyrphus grandicornis*, an Australasian species, has been anthropogenically introduced to Hawaii and French Polynesia where no previous hoverfly species were known to occur [15]. *Eristalis tenax* is thought to have been introduced to New Zealand from Britain or California some time before 1888 and is now very abundant in the country [93] and known to visit the flowers of various native plants [94]. The effects of these non-native species on the native flora and fauna are not currently known and this will be an important area of research to determine the effects of using some species as managed pollinators.

## 5. Pollinator declines, phenological shifts and threats

We are in an era of accelerating biodiversity loss, with particular concern over pollinator declines, phenological mismatches and the consequences for food security [95]. Climate change has been shown to force species to higher elevations and latitudes and lead to shifts in phenology [96]. Data on range shifts in hoverflies are limited, but a recent study predicted losses of some species from lowland areas and gains in alpine regions in southern Europe [97]. Agriculture is predominantly clustered in lowland areas and so loss of these pollinators could have detrimental effects. Hoverfly species with high mobility and high reproductive rates, such as migratory species, might be predisposed to shift ranges, as has been seen in other insect migrants [98–100], and this may be particularly important for countering crop damage caused by poleward shifts in aphid pests [101].

Pollinator declines may also be linked to habitat destruction and degradation following agricultural intensification and urbanization [95]. A number of studies have addressed pollinator declines using historical data to investigate changes in species assemblages and landscape occupancy. Interestingly, there are often contrasting patterns between hoverflies and wild bees. For example, studies investigating pollinator groups across European countries pre- and post-1980 (UK, The Netherlands and Belgium) have shown overall declines in wild bee diversity, while hoverfly diversity has not changed in the UK and has increased in the Netherlands [102]. The greatest declines in bees and hoverflies were found in species with narrow habitat requirements, dietary specializations and, within the UK, those with only a single generation per year [102]. By contrast, migratory hoverfly species with wide habitat ranges and multiple generations per year fared better [102]. Although this study did not measure population densities, shifts in the relative number of records suggest an increase in the domination of pollinator communities by a smaller number of species [102].

An analysis by Carvalheiro *et al.* [103] showed losses of butterfly and wild bee species richness in Britain, The Netherlands and Belgium pre-1990 (1930–1990), but with a slowing of this negative trend since [103]. By contrast, pre-1990, hoverflies showed no significant declines in species richness in any country and slight increases in the Netherlands, albeit with increases in homogeneity across space [103]. Post-1990, hoverfly species richness increased in Belgium with no significant changes in the UK or The Netherlands, while spatial homogenization essentially stopped for this group. The authors suggest

that this may be due to a period of increased conservation investment (post-1990), highlighting the potential to maintain or even restore current species assemblages in some areas.

Powney *et al.* [104] used data from 1980 to 2013 from Britain to reveal widespread variation in the landscape occupancy trends of wild pollinators: a third of pollinator species have decreased and a tenth increased, a trend that is shared between bees and hoverflies [104]. While the most severe declines in occupancy of wild bees were seen post-2007, hoverflies declined steadily from 1987 to 2012 but with little temporal variation in species evenness [104]. Using data produced in Powney *et al.* [104], we analysed trends in different larval guilds and found that those hoverfly species with larvae that develop in cow-dung (e.g. *Rhingia* spp.), tree sap (*Brachyopa*, *Chalcosyrphus*, *Ferdinandea* and *Sphegina* spp.), and in the nests of social, flying Hymenoptera (wasps and bees) (*Volucella* spp.) have all increased their occupancy during this period (1980–2013). The increase in species associated with dung seems to be due to a range expansion in the rare *Rhingia rostrata* (11.6% increase per year). For those that develop in sap runs, a single species, *Sphegina sibirica*, has expanded its occupancy dramatically (increasing by over 10% per year) and is becoming increasingly abundant. *Volucella* spp. have increased their occupancy with *V. inanis* (4.8%) and *V. zonaria* (5.6% per year) having well-documented range expansions [14]. The widespread migratory species *E. balteatus* and *E. tenax* also showed increased occupancy of 1.15% and 0.631% per year, respectively.

Research addressing changes in biomass for hoverflies is less common. Hallmann [105] compared abundance and richness of hoverflies at six locations in a German nature reserve in 1989 and 2014 and found an 80% decline in abundance, with almost all species showing numerical declines, and 20% declines in species richness [105]. Although this pattern is worrying, more evidence, particularly long-term and across multiple sites are needed to account for the large spatial and temporal variability in abundance seen in natural populations (figure 2*b*). For example, an analysis of 30 years (1973–2002) of suction trap data from four sites in Britain identified declines in overall insect biomass at only one site [106], while analysis of migratory hoverflies over southern Britain (2000–2009) revealed no population trend in total numbers [16]. An important limitation to these studies is a lack of data accounting for the first phase of agricultural intensification [106], meaning that shifting baselines may hide major population changes during this period, something that needs to be carefully considered when interpreting long-term trends in insect populations [107].

## 6. Conclusion and future research

There is growing appreciation of non-bee insects as key pollinators in many ecosystems [4,8]. Recent investigation into population trends of pollinating insects tend to show that hoverflies are declining to a lesser extent than other groups, such as wild bees [102–104], which may provide some level of robustness to pollination services in the face of environmental change. However, there is dire need to investigate these population trends over longer timescales and a broad range of sites in order to carefully disentangle the effects on different taxa and guilds. Anthropogenic landscape change such as urbanization and agricultural intensification may result

in communities being dominated by more generalist species, mobile migratory species [108], or change overwintering dynamics [109,110] and as yet we have little appreciation for how this may influence the stability of pollinator communities and the pollination functions they provide. As such, there is a need for further research to determine the response of hoverfly communities to anthropogenic perturbation. Furthermore, while hoverflies are recognized as important pollinators in many landscapes, much of the information is derived from temperate or Mediterranean regions and future research should aim to address this imbalance.

Unlike many other pollinator species, migratory hoverflies have the potential to transport pollen across vast distances and connect otherwise isolated plant populations, providing connectivity in disturbed and fragmented landscapes [16,73]. However, there is still much to be answered regarding the extent of this phenomenon, the routes taken, the resources required and the interspecific interactions that take place *en route* [111]. Hoverfly migration is likely to be much more common than currently recognized and is a rich area for further investigation, in particular with regards to encouraging and maintaining highly mobile pollinators into agroecosystems.

Hoverflies show immense potential as alternative managed pollinators, while also providing additional ecosystem services such as biological control of insect pests [112] and decomposition, an area that is strongly deserving of further exploration. While hoverfly pollination often falls short of many managed bees in terms of efficiency, it is noteworthy that commercial utilization of hoverflies has seen a steady rise in recent years with many companies supplying or investing in hoverfly systems to improve pollination, pest control and nutrient decomposition. Major challenges for sustained commercial pollination include optimization of rearing techniques, a better understanding of chemical and visual attractants and the effects of agrochemicals on hoverfly populations. In addition, the high mobility of many hoverfly species provides a significant challenge outside of closed greenhouse systems. Future research should address the timing and extent of foraging movements, both daily and through the life course, and the consequences of environmentally induced plasticity for movement ecology.

Data accessibility. This article has no additional data.

Competing interests. We declare we have no competing interests.

Funding. This work was supported through grants to K.R.W. from the Royal Society University Research Fellowship scheme (grant no. UF150126). T.D. and W.L.S.H. were supported by awards to K.R.W. from the Royal Society: a Fellows Enhancement Award (RGF\EA\ 180083) and a Research Grant for Research Fellows (RGF\R1\ 180047), respectively. R.M. was supported through the NERC GW4+ Doctoral Training Partnership (grant no. NE/L002434/1). Support to M.H.M.M. was through the European Union's Horizon 2020 research and innovation programme under the Marie Skłodowska-Curie grant agreement no. 795568.

Acknowledgements. We thank Dr Alison Scott-Brown for suggesting we review this topic, Dr Ximo Mengual and Dr Sergey Lysenkov for discussions on the classification of the family Syrphidae and the foraging ranges hoverflies, respectively, and the referees for their constructive comments.

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
