## [Reviewer comments · Proceedings of the Royal Society B: Biological Sciences]

Review History

RSPB-2020-0508.R0 (Original submission)

Review form: Reviewer 1

Recommendation

Accept with minor revision (please list in comments)

Scientific importance: Is the manuscript an original and important contribution to its field?

Good

General interest: Is the paper of sufficient general interest?

Good

Quality of the paper: Is the overall quality of the paper suitable?

Good

Is the length of the paper justified?

Yes

Should the paper be seen by a specialist statistical reviewer?

No

Do you have any concerns about statistical analyses in this paper? If so, please specify them explicitly in your report.

No

It is a condition of publication that authors make their supporting data, code and materials available - either as supplementary material or hosted in an external repository. Please rate, if applicable, the supporting data on the following criteria.

Is it accessible?

Yes

Is it clear?

Yes

Is it adequate?

Yes

Do you have any ethical concerns with this paper?

No

Comments to the Author

Present study is a well summarized review on the importance of hoverflies as non-bee pollinators and their relationship in the landscape (in terms of ecosystem services) with other bee pollinators. The sections are well categorized and overall the manuscript is well written. I also find the authors' efforts commendable in summing up the taxonomic (especially the morphological) diversity of hoverflies with landscape characteristics and ecosystem services.

My minor suggestions are:

1. While hoverflies are important pollinators in a landscape, the authors should also discuss more the pitfalls in such pollination services. Wildflowers and multiple crop/non-crop flowering patches certainly benefit from hoverflies as much as from bee pollinators. However, for sustained commercial pollination, the need for mobile hives (honey bee hives or bumble bee boxes for greenhouse pollination services) or alternate resources (alkali bees for example) are very important. What are the future directions for hoverflies in terms of management?
2. Write honeybees as honey bees and bumblebees as bumble bees. They are true bees. Hence the words must be separated.
3. L88 A reference for the link to the data.
4. L122-L124 Please rephrase for clarity.

Review form: Reviewer 2

Recommendation

Major revision is needed (please make suggestions in comments)

Scientific importance: Is the manuscript an original and important contribution to its field?

Acceptable

General interest: Is the paper of sufficient general interest?

Good

Quality of the paper: Is the overall quality of the paper suitable?

Acceptable

Is the length of the paper justified?

Yes

Should the paper be seen by a specialist statistical reviewer?

No

Do you have any concerns about statistical analyses in this paper? If so, please specify them explicitly in your report.

No

It is a condition of publication that authors make their supporting data, code and materials available - either as supplementary material or hosted in an external repository. Please rate, if applicable, the supporting data on the following criteria.

Is it accessible?

Yes

Is it clear?

Yes

Is it adequate?

Yes

Do you have any ethical concerns with this paper?

No

Comments to the Author

The manuscript entitled "Pollination by Hoverflies in the anthropocene" provides a review highlighting the importance of hoverflies, based on the fact that beside pollination, hoverflies provide ecosystem functions not seen in bees, such as crop protection from pests, recycling of organic matter and long-distance pollen transfer. They refer to research stating that migratory species in particular can be hugely abundant and unlike many insect pollinators, do not yet appear to be in serious decline. The authors contrast the roles of hoverflies and bees as pollinators, discuss the need for research and monitoring of different pollinator responses to anthropogenic change and examine emerging research into large populations of migratory hoverflies, the threats they face and how they might be utilised to improve sustainable agriculture.

General remarks:

As can be seen already from the abstract the title is very misleading and needs to be changed. Unfortunately, the authors cover much more than just pollination, most likely because not that much is known about syrphid pollination efficiency. I strongly disagree with the authors' apparent view that visitation and pollination are the same thing. Even pollen transport (and even deposition on stigma!) is no guarantee for pollination efficiency. The only sure test of pollination efficiency is to expose virgin flowers to single visits of potential pollinators and measure seed set. To my knowledge these experiments are rare in non-bee flower visitors (a notable exception is Fontaine et al. (2005), however not conducted in a natural setting). I urge the authors to make this clearer so as not to confuse the readers.

One of the main reasons I agreed to review this manuscript was that I thought it was about pollination as the title suggests. I feel that it would have been better to narrowly stick to pollination rather than to dabble into all sorts of syrphid natural history information, conservation issues and non-pollination ecosystem services provided, and naturally, due to space constraints, covering those subjects relatively half-heartedly.

I suggest the authors rethink the scope of the review and change the title and/or content accordingly.

Specific remarks:

Figure 2A: what is the measurement unit of distance here? I assume it's km but it should be written on the y-axis.

Decision letter (RSPB-2020-0508.R0)

03-Apr-2020

Dear Dr Wotton:

Your manuscript has now been peer reviewed and I've read through the manuscript myself. The reviewers' comments (not including confidential comments to the Editor) are included at the end of this email for your reference. As you will see, the reviewers think that the manuscript has potential, but have raised some concerns and so I would like to invite you to revise your manuscript to address them. The reviewers have different views on the inclusion of wider information (than pollination) about hoverflies -- referee 1 liked the broader context while referee 2 thinks it's a distraction. On balance, as *Proceedings B* is a general biology journal and many readers won't know that much about hoverflies, I think it's fair to include this material. However, the title might need to be tweaked to avoid disappointing readers like referee 2.

We do not allow multiple rounds of revision so we urge you to make every effort to fully address all of the comments at this stage. If deemed necessary, your manuscript will be sent back to one or more of the original reviewers for assessment. If the original reviewers are not available we may invite new reviewers. Please note that we cannot guarantee eventual acceptance of your manuscript at this stage.

Research ethics:

Use of animals and field studies:

If your study uses animals please include details in the methods section of any approval and licences given to carry out the study and include full details of how animal welfare standards were ensured. Field studies should be conducted in accordance with local legislation; please

include details of the appropriate permission and licences that you obtained to carry out the field work.

Please submit a copy of your revised paper within three weeks. If we do not hear from you within this time your manuscript will be rejected. If you are unable to meet this deadline please let us know as soon as possible, as we may be able to grant a short extension.

Best wishes,
Innes

Prof. Innes Cuthill
Reviews Editor, Proceedings B
<mailto:proceedingsb@royalsociety.org>

Reviewer(s)' Comments to Author:

Referee: 1

Comments to the Author(s)

Present study is a well summarized review on the importance of hoverflies as non-bee pollinators and their relationship in the landscape (in terms of ecosystem services) with other bee pollinators. The sections are well categorized and overall the manuscript is well written. I also find the authors' efforts commendable in summing up the taxonomic (especially the morphological) diversity of hoverflies with landscape characteristics and ecosystem services.

My minor suggestions are:

1. While hoverflies are important pollinators in a landscape, the authors should also discuss more the pitfalls in such pollination services. Wildflowers and multiple crop/non-crop flowering patches certainly benefit from hoverflies as much as from bee pollinators. However, for sustained commercial pollination, the need for mobile hives (honey bee hives or bumble bee boxes for greenhouse pollination services) or alternate resources (alkali bees for example) are very important. What are the future directions for hoverflies in terms of management?
2. Write honeybees as honey bees and bumblebees as bumble bees. They are true bees. Hence the words must be separated.
3. L88 A reference for the link to the data.
4. L122-L124 Please rephrase for clarity.

Referee: 2

Comments to the Author(s)

The manuscript entitled "Pollination by Hoverflies in the anthropocene" provides a review highlighting the importance of hoverflies, based on the fact that beside pollination, hoverflies provide ecosystem functions not seen in bees, such as crop protection from pests, recycling of organic matter and long-distance pollen transfer. They refer to research stating that migratory species in particular can be hugely abundant and unlike many insect pollinators, do not yet appear to be in serious decline. The authors contrast the roles of hoverflies and bees as pollinators, discuss the need for research and monitoring of different pollinator responses to anthropogenic change and examine emerging research into large populations of migratory hoverflies, the threats they face and how they might be utilised to improve sustainable agriculture.

General remarks:

As can be seen already from the abstract the title is very misleading and needs to be changed. Unfortunately, the authors cover much more than just pollination, most likely because not that much is known about syrphid pollination efficiency. I strongly disagree with the authors' apparent view that visitation and pollination are the same thing. Even pollen transport (and even deposition on stigma!) is no guarantee for pollination efficiency. The only sure test of pollination efficiency is to expose virgin flowers to single visits of potential pollinators and measure seed set. To my knowledge these experiments are rare in non-bee flower visitors (a notable exception is Fontaine et al. (2005), however not conducted in a natural setting). I urge the authors to make this clearer so as not to confuse the readers.

One of the main reasons I agreed to review this manuscript was that I thought it was about pollination as the title suggests. I feel that it would have been better to narrowly stick to pollination rather than to dabble into all sorts of syrphid natural history information, conservation issues and non-pollination ecosystem services provided, and naturally, due to space constraints, covering those subjects relatively half-heartedly.

I suggest the authors rethink the scope of the review and change the title and/or content accordingly.

Specific remarks:

Figure 2A: what is the measurement unit of distance here? I assume it's km but it should be written on the y-axis.

Author's Response to Decision Letter for (RSPB-2020-0508.R0)

See Appendix A.

Decision letter (RSPB-2020-0508.R1)

21-Apr-2020

Dear Dr Wotton

I am pleased to inform you that your manuscript entitled "Pollination by Hoverflies in the anthropocene" has been accepted for publication in Proceedings B.

If you are likely to be away from e-mail contact during this period, let us know. Due to rapid publication and an extremely tight schedule, if comments are not received, we may publish the paper as it stands.

Open access

You are invited to opt for open access via our author pays publishing model. Payment of open access fees will enable your article to be made freely available via the Royal Society website as soon as it is ready for publication. For more information about open access publishing please visit our website at http://royalsocietypublishing.org/site/authors/open_access.xhtml.

The open access fee is £1,700 per article (plus VAT for authors within the EU). If you wish to opt for open access then please let us know as soon as possible.

Paper charges

All supplementary materials accompanying an accepted article will be treated as in their final form. They will be published alongside the paper on the journal website and posted on the online

figshare repository. Files on figshare will be made available approximately one week before the accompanying article so that the supplementary material can be attributed a unique DOI.

Sincerely,

Proceedings B
mailto: proceedingsb@royalsociety.org

Appendix A

Centre for Ecology
and Conservation

Centre for Ecology and Conservation
University of Exeter
Penryn Campus,
Penryn, Cornwall
TR10 9FE
UK
t: +44 (0)1326 25 4118
e: k.r.wotton@exeter.ac.uk
w: www.exeter.ac.uk

14 May 2020

Dear Professor Cuthill,

We thank the referees for their feedback on our manuscript (RSPB-2020-0508). We have now revised our paper in line with the comments and below provide a point-by-point response.

We trust that you find our submission in order and look forward to hearing from you in due course.

Sincerely yours,

Dr Karl Wotton

Senior Lecturer & Royal Society University Research Fellow, University of Exeter, UK

Reply to referees. Original comments in black, our response in red.

Referee 1.

1. While hoverflies are important pollinators in a landscape, the authors should also discuss more the pitfalls in such pollination services. Wildflowers and multiple crop/non-crop flowering patches certainly benefit from hoverflies as much as from bee pollinators. However, for sustained commercial pollination, the need for mobile hives (honey bee hives or bumble bee boxes for greenhouse pollination services) or alternate resources (alkali bees for example) are very important. What are the future directions for hoverflies in terms of management?

We have added an additional section to our conclusions, it reads:

‘Hoverflies show immense potential as alternative managed pollinators, an area that is strongly deserving of further exploration. While hoverfly pollination often falls short of many managed bees in terms of efficiency, it is noteworthy that commercial utilisation of hoverflies has seen a steady rise in recent years with many companies supplying or investing in hoverfly systems to improve pollination, pest control and nutrient decomposition. Major challenges for sustained commercial pollination include optimisation of rearing techniques, a better understanding of chemical and visual attractants and the effects of agrochemicals on hoverfly populations. In addition, the high mobility of many hoverfly species provides a significant challenge outside of closed greenhouse systems. Future research should address the timing and extent of foraging movements, both daily and through the life course, and the consequences of environmentally induced plasticity for movement ecology.’

2. Write honeybees as honey bees and bumblebees as bumble bees. They are true bees. Hence the words must be separated.

We have made these changes as suggested.

3. L88 A reference for the link to the data.

We have added a reference to the Food and agriculture Organisation of the United Nations data website (FAOSTAT).

4. L122-L124 Please rephrase for clarity.

We have rephrased this section as suggested. It now reads:

‘In apple orchards, hairy bumble bee mimics like *Merodon equestris* (Figure 1F) can carry around 10,000 pollen grains (of which 29% was fruit pollen) while honey bee mimics such as *E. tenax* can carry around 3,500 (67% fruit pollen) [38]. In contrast, pollen loads of bumble bee *Bombus terrestris* and the honey bee *Apis mellifera* were 19,000 (85% fruit pollen) and 5,600 thousand (73% fruit pollen), respectively [38].’

Referee 2

1. The title is very misleading and needs to be changed. Unfortunately, the authors cover much more than just pollination, most likely because not that much is known about syrphid pollination efficiency.

Please see comment 3 below.

2. I strongly disagree with the authors' apparent view that visitation and pollination are the same thing. Even pollen transport (and even deposition on stigma!) is no guarantee for pollination efficiency. The only sure test of pollination efficiency is to expose virgin flowers to single visits of potential pollinators and measure seed set. To my knowledge these experiments are rare in non-bee flower visitors (a notable exception is Fontaine et al. (2005), however not conducted in a natural setting). I urge the authors to make this clearer so as not to confuse the readers.

We agree with referee 2 and this was not a point we intended to make. We have corrected the error where we wrongly used the term pollination. In addition, we now also draw attention to the lack of experimental approaches outlined by the referee to test pollination efficiency outside of non-bee flower visitors. This paragraph [L140] now begins:

'Pollinator efficiency, as measured by seed set following a single visit from a potential pollinator to a virgin flower, has seldom been quantified for non-bee flower visitors. However, seed set has been measured following multiple visits for various hoverfly species.'

3. One of the main reasons I agreed to review this manuscript was that I thought it was about pollination as the title suggests. I feel that it would have been better to narrowly stick to pollination rather than to dabble into all sorts of syrphid natural history information, conservation issues and non-pollination ecosystem services provided, and naturally, due to space constraints, covering those subjects relatively half-heartedly. I suggest the authors rethink the scope of the review and change the title and/or content accordingly.

We feel that our title reflects the core purpose of the review, that is to cover hoverfly pollination in a changing world. As such, natural history and conservation issues form an important part of the narrative and provide background to the reader that is not easily assessable nor common knowledge. We do not believe that this has been done 'relatively half-heartedly', for example we provide a complete synthesis of the current state of the knowledge on hoverfly declines, including novel data, that is relevant to the role of hoverflies as pollinators. In addition, we provide a unique appraisal of the importance of migratory hoverflies for pollination. However, we do provide a short standalone section on other ecosystem services which we have now removed while retaining the pertinent information and references within the main text. Given this content change, and the explanation set out above, we feel that the original title now more strongly reflects the core purpose of the review and would prefer to keep this version. Alternatively, we could live with 'Pollination and Ecology of Hoverflies in the anthropocene' if deemed necessary.

4. Figure 2A: what is the measurement unit of distance here? I assume it's km but it should be written on the y-axis.

The value is a relative distance calculated as a flight length divided by the mean distance between flowers (recording units). We have updated our axis label and figure legend to make this clearer, it now reads:

Flight distance constructed as regression lines of the relative distance (flight length / mean distance between recording units of flowers, inflorescences, or plants) plotted against the number of flights for hoverflies (blue lines), bees (orange lines) and butterflies (grey line). Redrawn from (Lysenkov 2009).

We have made a few minor additional changes that I outline below.

- We have added a missing citation for: Willmer, P. (2011). Pollination by Flies. In: Pollination and Floral Ecology. pp. 304–321.
- We have also added citations providing additional evidence of (1) larger foraging ranges in hoverflies than bees: Rader et al. 2011. Pollen transport differs among bees and flies in a human-modified landscape. *Divers. Distrib.*; and (2) for hoverfly migration in Australia: Finch & Cook 2020. Flies on vacation: evidence for the migration of Australian Syrphidae (Diptera). *Ecol. Entomol.*;
- We now cite two recent reviews on (1) interspecific interactions during co-migrations that is particularly relevant to hoverfly movement ecology: Cohen & Satterfield 2020. 'Chancing on a spectacle:' co-occurring animal migrations and interspecific interactions. *Ecography*; and (2) on insect declines: Didham et al. 2020. Interpreting Insect Declines: Seven Challenges and a Way Forward. *Insect Conservation and Diversity*.